# Reporting and misreporting of sex differences in the biological sciences

Yesenia Garcia-Sifuentes[1], Donna L Maney[1,2]*

[1]Graduate Program in Neuroscience, Emory University, Atlanta, United States;
[2]Department of Psychology, Emory University, Atlanta, United States

**Abstract** As part of an initiative to improve rigor and reproducibility in biomedical research, the U.S. National Institutes of Health now requires the consideration of sex as a biological variable in preclinical studies. This new policy has been interpreted by some as a call to compare males and females with each other. Researchers testing for sex differences may not be trained to do so, however, increasing risk for misinterpretation of results. Using a list of recently published articles curated by Woitowich et al. (eLife, 2020; 9:e56344), we examined reports of sex differences and non-differences across nine biological disciplines. Sex differences were claimed in the majority of the 147 articles we analyzed; however, statistical evidence supporting those differences was often missing. For example, when a sex-specific effect of a manipulation was claimed, authors usually had not tested statistically whether females and males responded differently. Thus, sex-specific effects may be over-reported. In contrast, we also encountered practices that could mask sex differences, such as pooling the sexes without first testing for a difference. Our findings support the need for continuing efforts to train researchers how to test for and report sex differences in order to promote rigor and reproducibility in biomedical research.

*For correspondence:
dmaney@emory.edu

**Competing interest:** The authors declare that no competing interests exist.

## Introduction

Historically, biomedical research has not considered sex as a biological variable (SABV). Including only one sex in preclinical studies—or not reporting sex at all—is a widespread issue (*Sugimoto et al., 2019*). In a cross-disciplinary, quantitative assessment of the 2009 biomedical literature, *Beery and Zucker, 2011*, found a concerning bias toward the use of males only. As awareness of this issue increased, in 2016 the National Institutes of Health (NIH) implemented a policy requiring consideration of SABV in the design, analysis, and reporting of all NIH-funded preclinical research (*NIH, 2015*; *Clayton, 2018*). By addressing the long-standing over-representation of male non-human animals and cells, the policy was intended not only to ameliorate health inequities but to improve rigor and reproducibility in biomedical research (*Clayton and Collins, 2014*). Since 2016, NIH has made available a number of resources, including training modules, administrative funding supplements, and a center program focused on sex differences (*Arnegard et al., 2020*). These efforts have resulted in the discovery of new sex differences across a wide spectrum of research fields (*Arnegard et al., 2020*).

Although the NIH policy does not explicitly require that males and females be compared directly with each other, the fact that more NIH-funded researchers must now study both sexes should lead to an increase in the frequency of such comparisons (*Maney, 2016*). For example, there should be more testing for sex-specific responses to experimental treatments. However, in a follow-up to *Beery and Zucker, 2011*, study, *Woitowich et al., 2020*, showed evidence to the contrary. Their analysis revealed that between 2011 and 2019, although the proportion of articles that included both sexes significantly increased (see also *Will et al., 2017*), the proportion that treated sex as a variable did not. This finding contrasts sharply with expectations, given not only the NIH mandate but also numerous calls over the past decade to disaggregate all preclinical data by sex and to test for sex differences

**eLife digest** Biomedical research has a long history of including only men or male laboratory animals in studies. To address this disparity, the United States National Institutes of Health (NIH) rolled out a policy in 2016 called Sex as a Biological Variable (or SABV). The policy requires researchers funded by the NIH to include males and females in every experiment unless there is a strong justification not to, such as studies of ovarian cancer. Since then, the number of research papers including both sexes has continued to grow.

Although the NIH does not require investigators to compare males and females, many researchers have interpreted the SABV policy as a call to do so. This has led to reports of sex differences that would otherwise have been unrecognized or ignored. However, researchers may not be trained on how best to test for sex differences in their data, and if the data are not analyzed appropriately this may lead to misleading interpretations.

Here, Garcia-Sifuentes and Maney have examined the methods of 147 papers published in 2019 that included both males and females. They discovered that more than half of these studies had reported sex differences, but these claims were not always backed by statistical evidence. Indeed, in a large majority (more than 70%) of the papers describing differences in how males and females responded to a treatment, the impact of the treatment was not actually statistically compared between the sexes. This suggests that sex-specific effects may be over-reported. In contrast, Garcia-Sifuentes and Maney also encountered instances where an effect may have been masked due to data from males and females being pooled together without testing for a difference first.

These findings reveal how easy it is to draw misleading conclusions from sex-based data. Garcia-Sifuentes and Maney hope their work raises awareness of this issue and encourages the development of more training materials for researchers.

(e.g., *Becker et al., 2016*; *Potluri et al., 2017*; *Shansky and Murphy, 2021*; *Tannenbaum et al., 2019*; *Woitowich and Woodruff, 2019*).

One potential barrier to SABV implementation is a lack of relevant resources; for example, not all researchers have received training in experimental design and data analysis that would allow them to test for sex differences using appropriate statistical approaches. This barrier is quite important not only because it prevents rigorous consideration of sex in the first place, but also because any less-than-rigorous test for sex differences creates risk for misinterpretation of results and dissemination of misinformation to other scientists and to the public (*Maney, 2016*). In other words, simply calling for the sexes to be compared is not enough if researchers are not trained to do so; if SABV is implemented haphazardly, it has the potential to decrease, rather than increase, rigor and reproducibility.

In this study, our goal was to analyze recently published articles to determine how often sex differences are being reported and what statistical evidence is most often used to support findings of difference. To conduct this assessment, we leveraged the collection of articles originally curated by *Woitowich et al., 2020*, for their analysis of the extent to which SABV is being implemented. Their original list, which was itself generated using criteria developed by *Beery and Zucker, 2011*, included 720 articles published in 2019 in 34 scholarly journals within nine biological disciplines. Of those, Woitowich et al. identified 151 articles that included females and males and that analyzed data disaggregated by sex or with sex as fixed factor or covariate. Working with that list of 151 articles, we asked the following questions for each: First, was a sex difference reported? If so, what statistical approaches were used to support the claim? We focused in particular on studies with factorial designs in which the authors reported that the effect of one factor, for example treatment, depended on sex. Next, we asked whether data from males and females were kept separate throughout the article, and if they were pooled, whether the authors tested for a sex difference before pooling. Finally, we noted whether the authors used the term 'sex' or 'gender', particularly in the context of preclinical (non-human animal) studies.

**Table 1.** Journals surveyed by discipline.
The categorization of journals into disciplines was as defined by *Beery and Zucker, 2011*, and *Woitowich et al., 2020*.

| Discipline | Journal 1 | Journal 2 | Journal 3 | Journal 4 | No. articles |
|---|---|---|---|---|---|
| Behavior | *Behavioral Ecology and Sociobiology* | *Animal Behavior* | *Animal Cognition* | *Behavioral Ecology* | 40 |
| Behavioral Physiology | *Journal of Comparative Psychology* | *Behavioral Neuroscience* | *Physiology and Behavior* | *Hormones and Behavior* | 20 |
| Endocrinology | *European Journal of Endocrinology* | *Journal of Neuroendocrinology* | *Endocrinology* | *American Journal of Physiology – Endocrinology and Metabolism* | 27 |
| General Biology | *PLoS Biology* | *Proceedings of the Royal Society B: Biological Sciences* | *Nature* | *Science* | 9 |
| Immunology | *Journal of Immunology* | *Infection and Immunity* | *Immunity* | *Vaccine* | 10 |
| Neuroscience | *Journal of Neuroscience* | *Neuroscience* | *Journal of Comparative Neurology* | *Nature Neuroscience* | 9 |
| Pharmacology | *Neuropsychopharmacology* | *Journal of Psychopharmacology* | *Journal of Pharmacology and Experimental Therapeutics* | *British Journal of Pharmacology* | 11 |
| Physiology | *Journal of Physiology (London)* | *American Journal of Physiology – Renal Physiology* | *American Journal of Physiology – Gastrointestinal and Liver Physiology* | *American Journal of Physiology – Heart and Circulatory Physiology* | 12 |
| Reproduction | *Biology of Reproduction* | *Reproduction* | | | 9 |

## Results

We began with 151 articles, published in 2019, that were determined by *Woitowich et al., 2020*, to have (1) included both males and females and (2) reported data by sex (disaggregated or with sex included in the statistical model). Of those, we identified four that contained data from only one sex (e.g., animals of the other sex had been used only as stimulus animals or to calculate sex ratios). After excluding those articles, our final sample size was 147. See *Table 1* for the sample sizes of articles from each discipline. More than one-third of the studies were on humans (35%) and a similarly large proportion on rats or mice (31%). The remainder encompassed a wide variety of species including non-human primates, dogs, cats, pigs, sheep, deer, squirrels, racoons, Tasmanian devils, lemur, lions, meerkats, and mongoose. The codes are given in *Supplementary file 1a*, and results of coding are given in *Supplementary file 1b* (individual articles) and *Supplementary file 1C* (tabulated data).

### Question 1: Was a sex difference reported?

Results pertaining to Question 1 are shown in *Figure 1A*. Comparing the sexes, either statistically or by assertion, was common, occurring in 80% of the articles. A positive finding of a sex difference was reported in 83 articles, or 57%. Of the articles reporting a sex difference, 41 (49% of the 83 articles) mentioned that result in the title or the abstract. Thus, in our sample of articles in which data were reported by sex, a sex difference was reported in more than half of the articles and in half of those, the difference was treated as a major finding by highlighting it in the title or abstract. In 44% of articles, a sex difference was neither stated nor implied.

These results are broken down by discipline in *Figure 1B*. The sexes were most commonly compared in the field of Endocrinology (93%) and least often in the field of Neuroscience (33%). In the field of Reproduction, the sexes were compared 89% of the time and in 100% of those cases, a sex difference was mentioned in the title or abstract. Sex differences were least likely to be emphasized in the title or abstract in the fields of General Biology and Neuroscience (11% each).

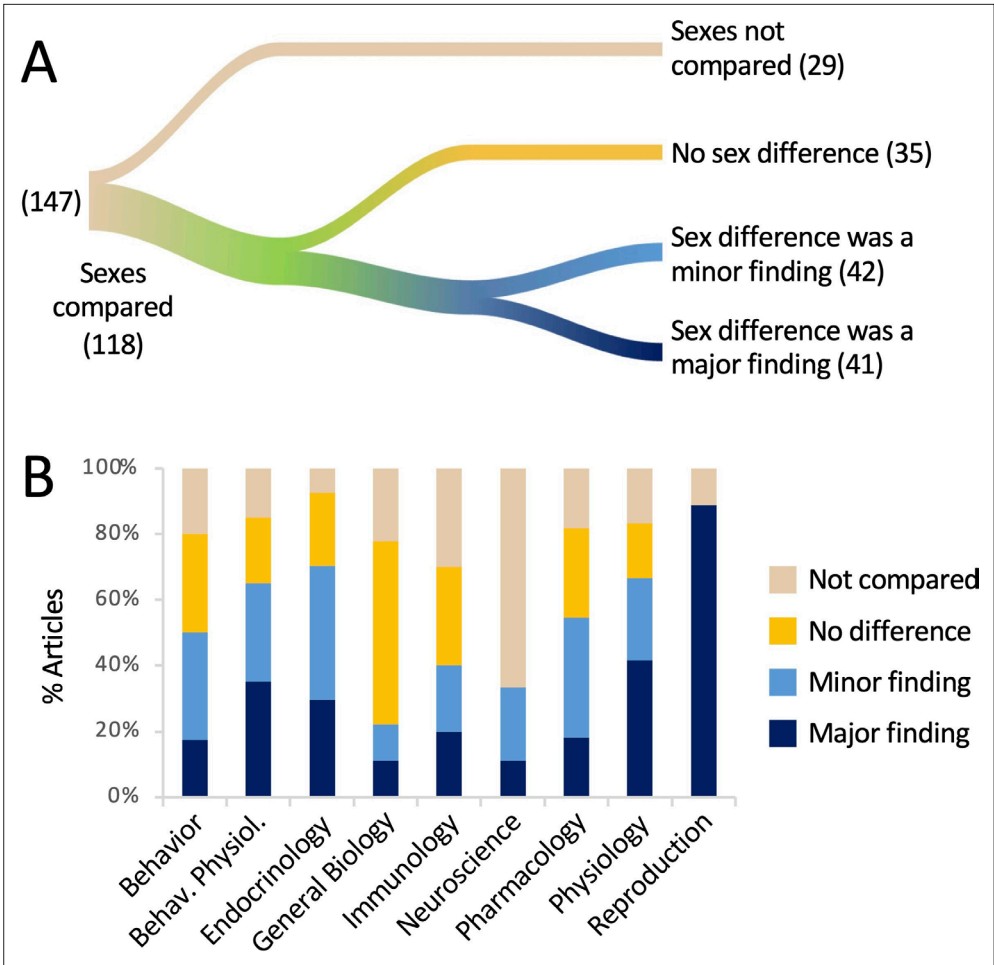

**Figure 1.** The sexes were compared in the majority of the articles analyzed. (**A**) The river plot shows the proportions of articles comparing the sexes, either statistically or qualitatively, and the outcomes of those comparisons. The width of each stream is proportional to the number of articles represented in that stream. The numbers of articles are given in parentheses. If a sex difference was mentioned in the title or abstract, the article was coded as 'major finding'. For a river plot showing how (**A**) fits into the larger context of the study by *Woitowich et al., 2020*, please see *Figure 1—figure supplement 1*. (**B**) The percentage of articles in which sexes were compared is plotted for each discipline. All data are shown in *Supplementary file 1* and *Figure 1—source data 1*.

The online version of this article includes the following figure supplement(s) for figure 1:

**Source data 1.** Data depicted in *Figure 1*.

**Figure supplement 1.** River plot showing our findings in the larger context of the study by *Woitowich et al., 2020*.

Although a sex difference was claimed in a majority of articles (57%), not all of these differences were supported with statistical evidence. In more than a quarter of the articles reporting a sex difference, or 24/83 articles, the sexes were never actually compared statistically. In these cases, the authors claimed that the sexes responded differentially to a treatment when the effect of treatment was not statistically compared across sex. This issue is explored in more detail under *Question 2*, below. Finally, we noted at least five articles in which the authors claimed that there was no sex difference, but did not appear to have tested statistically for one.

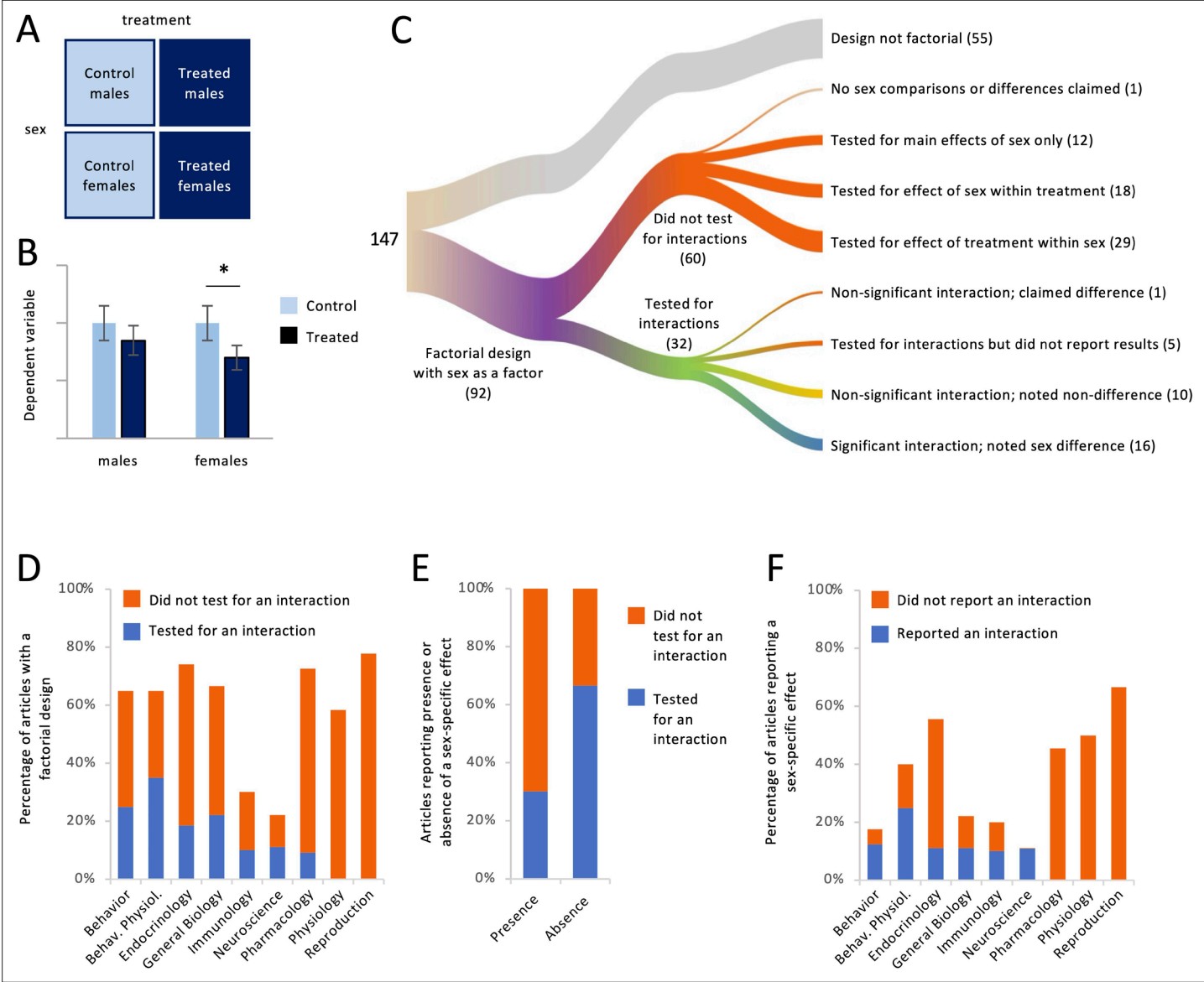

**Figure 2.** Factorial designs and sex-specific effects. For each article, we noted whether it contained a study with a factorial design with sex as a factor (**A**), for example, males and females nested inside treated and control groups. (**B**) In this hypothetical dataset, there was a significant effect of treatment only in females. Some authors would claim that the treatment had a 'sex-specific' effect without testing statistically whether the response to treatment depended on sex. In this example, it does not (see *Maney, 2016*; *Nieuwenhuis et al., 2011*). (**C**) The river plot shows the proportion of articles with a factorial design and the analysis strategy for those. The width of each stream is proportional to the number of articles represented in that stream. The numbers of articles are given in parentheses. (**D**) The percentage of articles with a factorial design (see A) is plotted for each discipline. Only a minority tested for an interaction between sex and other factors. (**E**) Testing for an interaction was less common in articles claiming the presence of a sex-specific effect, meaning a sex difference in the degree to which a second variable affected the outcome variable, than in articles claiming the absence of such an effect. (**F**) The percentage of articles claiming a sex-specific effect is plotted for each discipline. Only a minority reported a significant interaction.

The online version of this article includes the following source data for figure 2:

**Source data 1.** Data depicted in *Figure 2*.

## Question 2: Did the study have a factorial design with sex as a factor, and if so, did the authors test statistically whether the effect of other factors depended on sex?

For each article, we asked whether it contained a study with a factorial design in which sex was one of the factors. This design is common when researchers are interested in testing whether the sexes

respond differently to a manipulation such as a drug treatment (*Figure 2A*). Below, we use the term 'treatment' to refer to any non-sex factor in a factorial design. Such factors were not limited to treatment, however; they also included variables such as genotype, season, age, exposure to stimuli, etc. Hypothetical results of a study with such a design are shown in *Figure 2B*. In order to draw a conclusion about whether responses to treatment differed between females and males, the effect of the treatment must be compared across sex. Although there are several ways of making such a comparison (see *Cumming, 2012*; *Gelman and Stern, 2006*), it is typically done by testing for an interaction between sex and treatment. If the interaction is significant, then a claim can be made that the sexes responded differently to the treatment. Comparing the treated and control groups within each sex, in other words disaggregating the data by sex and testing for effects of treatment separately in females and males, does not test whether the sexes responded differently; that is, it does not test whether the magnitude of the response differs between females and males (*Gelman and Stern, 2006*; *Makin and Orban de Xivry, 2019*; *Maney, 2016*; *Nieuwenhuis et al., 2011*; *Radke et al., 2021*).

The results pertaining to Question 2 are shown in *Figure 2C-F*. Out of the 147 articles we analyzed, 92 (63%) contained at least one study with a factorial design in which sex was a factor (*Figure 2C*). Regardless of whether a sex difference was claimed, we found that the authors explicitly tested for interactions between sex and other factors in only 27 of the 92 articles (29%). That is, authors tested statistically for a sex difference in the responses to other factor(s) less than one-third of the time. Testing for interactions with sex varied by discipline (*Figure 2D*). Authors were most likely to test for and report the results of interactions in the field of Behavioral Physiology (54% of relevant articles) and least likely in the fields of Physiology (0%) and Reproduction (0%).

Of the studies with a factorial design, 58% reported that the sexes responded differently to one or more other factors. The language used to state these conclusions often included the phrase 'sex difference' but could also include 'sex-specific effect' or that a treatment had an effect 'in males but not females' or vice versa. Of the 53 articles containing such conclusions, the authors presented statistics showing a significant interaction, in other words appropriate evidence that females and males responded differently, in only 16 (30%; *Figure 2E*, blue color in first column). In an additional article, the authors presented statistical evidence that the interaction was non-significant, yet claimed a sex-specific effect nonetheless. In five other articles, the authors mentioned testing for interactions but presented no results or statistics (e.g., p values) for those interactions. In the remainder of articles containing claims of sex-specific effects, the authors took one of two approaches; neither approach included testing for interactions. Instead, authors proceeded to what would normally be the post hoc tests conducted after finding a significant interaction. In 24 articles (45% of articles with claims of sex-specific effects), authors reported the effect of treatment within each sex and, reaching different conclusions for each sex (e.g., finding a p value below 0.05 in one sex but not the other), inappropriately argued that the response to treatment differed between females and males (see *Figure 2B*). In seven other articles claiming a sex-specific effect (13%), the sexes were compared within treatment; for example, authors compared the treated males with the treated females, not considering the control animals. Neither approach tests whether the treatment had different effects in females and males. Thus, a substantial majority of articles containing claims of sex-specific effects (70%) did not present statistical evidence to support those claims (*Figure 2E*, red color in first column); further, in the majority of articles without such evidence (24/37), the sexes were never compared statistically at all.

The omission of tests for interactions was related to whether researchers were claiming sex differences or not. Among the articles that were missing tests for interactions and yet contained conclusions about the presence or absence of sex-specific effects (41 articles), those claims were in favor of sex differences 88% of the time, compared with only 12% claiming that the responses in females and males were similar. Of all of the articles claiming similar responses to treatment, authors tested for interactions in the majority of cases (67%; *Figure 2E* blue color in second column).

The prevalence of reporting sex-specific effects is broken down by discipline in *Figure 2F*. The field with the lowest percentage of sex-specific effects was Behavior (18%), and that field also had the highest rate of backing up such claims with statistical evidence (71%). The field most likely to contain claims of sex-specific effects was Reproduction (67%), but this field was among three for which such claims were never backed up with statistical evidence (0% for Reproduction, Physiology, or Pharmacology).

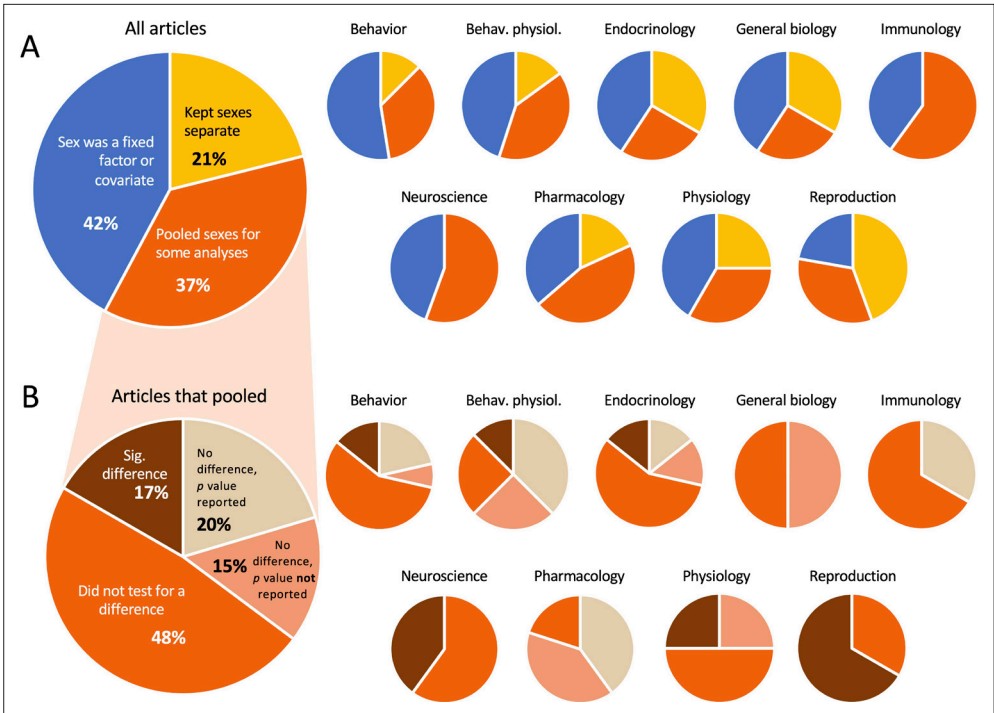

**Figure 3.** Proportion of articles in which the sexes were pooled. (**A**) In our sample, roughly one-third of the articles pooled the sexes for at least some analyses. (**B**) Among the articles that pooled, more than half did not test for a sex difference before pooling. In both (**A**) and (**B**), the smaller pie charts show the proportions within discipline. For the data used to make the charts, see *Supplementary file 1c* and *Figure 3—source data 1*.

The online version of this article includes the following source data for figure 3:

**Source data 1.** Data depicted in *Figure 3*.

## Question 3: Were the data from males and females pooled for any of the analyses?

In this study we included only articles in which data were reported by sex as previously determined by *Woitowich et al., 2020*. Thus, any articles in which the sexes were pooled for all analyses were not included here. We assigned each of the 147 articles to one of three categories, as follows (*Figure 3A*). In 31 (21%) of the articles, data from males and females were analyzed separately throughout. In 62 (42%) of the articles, males and females were analyzed in the same statistical models, but in those cases sex was included as a fixed factor or a covariate. In most cases when sex was a covariate, authors reported the results of the effect of sex rather than simply controlling for sex. In the remaining 54 (37%) articles, the sexes were pooled for at least some of the analyses.

Among the articles in which the sexes were pooled, the authors did so without testing for a sex difference almost half of the time (48%; *Figure 3B*). When authors did test for a sex difference before pooling, they sometimes found a significant difference yet pooled the sexes anyway; this occurred in 17% of the articles that pooled. When the sexes were pooled after finding no significant difference (35% of the articles that pooled), authors presented p values for the sex difference the majority of the time (11 out of 19 articles). Those p values ranged from 0.15 to >0.999. We noted no effect sizes reported in the context of pooling.

Across disciplines, pooling was most prevalent in Immunology (60%) and least prevalent in General Biology (22%). Males and females were most likely to be kept separate in General Biology (56%) and most likely to be included in statistical models in the field of Behavior (53%). When females and males were pooled, authors in the field of Immunology were least likely to have tested for a sex difference before pooling (33%) and most likely to do so in Pharmacology (80%). Pooling after finding a significant difference was most common in the field of Reproduction (67% of articles that pooled).

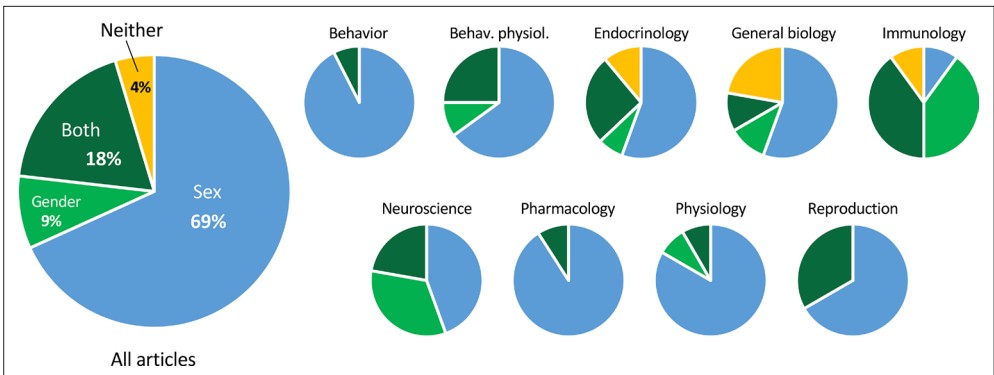

**Figure 4.** Proportions of articles using the terms 'sex' and 'gender'. The smaller pie charts show the proportions within discipline. The charts include all articles analyzed, on both humans and non-humans. For the data used to make the charts, see *Supplementary file 1C* and *Figure 4—source data 1*.

The online version of this article includes the following source data for figure 4:

**Source data 1.** Data depicted in *Figure 4*.

## Question 4: Was the term 'gender' used for non-human animals?

To refer to the categorical variable comprising male/female or man/woman (all were binary), the term 'sex' was used exclusively in 69% of the articles (*Figure 4*). 'Gender' was used exclusively in 9%, and both 'sex' and 'gender' were used in 19%. When both terms were used, they usually seemed to be used interchangeably. In 4% of the articles, neither term was used.

Of the articles in which the term 'gender' was used, 20% of the time it referred to non-human animals, such as mice, rats, and pigs. In one case, both 'sex' and 'gender' were used to refer to non-human animals in the title. In another case, 'gender' was used to refer to human cells. The majority of articles on non-human species used 'sex' (85%).

## Discussion

### Finding sex differences

*Woitowich et al., 2020*, found that over the past decade, the proportion of biological studies that included both females and males has increased, but the proportion in which sex is treated as a variable has not. Here, we have taken a closer look at the studies determined by those authors to have reported data by sex, that is, to have conformed to NIH guidelines on SABV. We found that in this subset of studies, authors typically also compared the sexes either statistically or by assertion (>80% of cases). Thus, the authors that complied with NIH guidelines to disaggregate data usually went *beyond* NIH guidelines to explicitly compare the sexes with each other. This finding is consistent with a larger analysis of articles in the field of Neuroscience from 2010 to 2014; when authors disaggregated data by sex, they usually proceeded to compare the sexes as well (*Will et al., 2017*). It is important to note, however, that both *Will et al., 2017*, and *Woitowich et al., 2020*, found that data were not analyzed by sex in the majority of articles that included both sexes (see *Figure 1—figure supplement 1*). Thus, our current finding that the sexes were usually compared should be interpreted in the context of the subset of articles following NIH guidelines. In the set of articles analyzed here, sex differences were claimed in a majority and were often highlighted in the title or abstract. We therefore found little evidence that researchers—at least those who comply with NIH guidelines—are uninterested in sex differences. We cannot rule out the possibility, however, that the researchers following NIH guidelines are primarily those that are interested in sex differences.

### Testing for interactions in a factorial design

Testing whether the sexes respond differently to a treatment requires statistical comparison between the two effects, which is typically done by testing for a sex × treatment interaction. In our analysis, however, tests for interactions were done only 29% of the time (*Figure 2C and D*). In the remaining 71%, the most common method for detecting differential effects of treatment was to compare

qualitatively the conclusions drawn for each sex; that is, to assert that a p value below 0.05 for one sex but not the other (*Figure 2B*) represents a meaningful difference between the effects. But null hypothesis significance testing does not allow for such conclusions (*Cumming, 2012*). This error, and the frequency with which it is made, has been covered in multiple publications; for example *Gelman and Stern, 2006*, titled their commentary "The difference between 'significant' and 'not significant' is not itself statistically significant." *Makin and Orban de Xivry, 2019*, included the error in their 'Top ten list of common statistical mistakes'. In an analysis of 520 articles in the field of Neuroscience, *Nieuwenhuis et al., 2011*, found that the error was committed in about half of articles containing a factorial design. The current analysis showed that, even a decade later, the frequency of this error in the field of Neuroscience has not changed (*Figure 2D*), at least when sex is one of the factors under consideration. The frequency of the error was high in most of the other disciplines as well, particularly Physiology and Reproduction, for which we found that authors never tested for interactions even though doing so was necessary to test their hypotheses about sex.

Statements such as the following, usually made without statistical evidence, were common: 'The treatment increased expression of gene X in a sex-dependent manner'; 'Our results demonstrate that deletion of gene X produces a male-specific increase in the behavior'; 'Our findings indicate that females are more sensitive to the drug than males'. In some of these cases, the terms 'sex-specific', 'sex-dependent', or 'sexual dimorphism' were used in the title of the article despite a lack of statistical evidence supporting the claim. In many of these articles, some of which stated that finding a sex difference was the major goal of the study, the sexes were not statistically compared at all. Thus, a lack of statistical evidence for sex-specific effects did not prevent authors from asserting such effects. Authors failing to test for interactions were far more likely to claim sex-specific effects than not (88% vs. 12%; *Supplementary file 1c*); they were also more likely to do so than were authors that did test for interactions (88% vs. 63%; *Supplementary file 1c*). Statistical analysis of these data showed that, in fact, sex-specific effects were reported significantly more often when no tests for interactions were reported ($\chi^2$ = 5.84; p = 0.016). Together, these results suggest a bias toward finding sex differences. In the absence of evidence, differences were claimed more often than not. A bias toward finding sex differences, where there are none, could artificially inflate the importance of sex in the reporting of biological data. Given that findings of sex × treatment interactions are rare in the human clinical literature, with false positives outnumbering false negatives (*Wallach et al., 2016*), and given also that sex differences are often reported in the media and used to shape education and health policy (*Maney, 2014*), it is especially important to base conclusions from preclinical research on solid statistical evidence.

## Pooling across sex

The set of articles we analyzed was pre-screened by *Woitowich et al., 2020*, to include only studies in which sex was considered as a variable. Nonetheless, even in this sample, data were often pooled across sex for some of the analyses (*Figure 3A*). In a majority of these articles, authors did not test for a sex difference before pooling (*Figure 3B*). Thus, for at least some analyses represented here, the data were not disaggregated by sex, sex was not a factor in those analyses, and we do not know whether there might have been a sex difference. Even when authors did test for a sex difference before pooling, the relevant statistics were often not presented. Finding and reporting a significant sex difference did not seem to reduce the likelihood that the sexes would be pooled. Note that the original sample of 720 articles in the study by Woitowich et al. included 251 articles in which sex was either not specified or the sexes were pooled for all analyses (*Figure 1—figure supplement 1*). Thus, the issue is more widespread than is represented in the current study. Pooling is not consistent with the NIH mandate to disaggregate data by sex and can prevent detection of meaningful differences. We note further that effect sizes were generally not reported before pooling; in addition to p values, effect sizes would be valuable for any assessment of whether data from males and females can be pooled without masking a potentially important difference (*Beltz et al., 2019*; *Diester et al., 2019*).

## Correcting for multiple comparisons

In their article on 'Ten statistical mistakes…,' *Makin and Orban de Xivry, 2019*, list another issue that is likely to be relevant to the study of sex differences: comparing multiple dependent variables across sex without correcting for multiple comparisons. The omission of such a correction increases the risk

of false positives, that is, making a type I error, which would result in over-reporting of significant effects. This risk is particularly important for researchers trying to comply with SABV, who may feel compelled to test for sex differences in every measured variable. In the current study, we found this issue to be prevalent. For example, we noted articles in which researchers measured expression of multiple genes in multiple tissues at multiple time points, resulting in a large number of comparisons across sex. In one such study, authors made 90 separate comparisons in the same set of animals and found five significant differences, which is exactly the number one would expect to find by chance. Although opinions vary about when corrections are necessary, omitting them when they are clearly needed is likely contributing to over-reporting of sex differences broadly across disciplines.

## Usage of 'sex' and 'gender'

We found that a large majority of studies on non-human animals used 'sex' to refer to the categorical variable comprising females and males. In eight articles, we noted usage of the word 'gender' for non-human animals. This usage appears to conflict with current recommendations regarding usage of 'gender', that is, gender should refer to socially constructed identities or behaviors rather than biological attributes (*Clayton and Tannenbaum, 2016*; *Holmes and Monks, 2019*; *Woitowich and Woodruff, 2019*). We did not, however, investigate the authors' intended meaning of either term. Although definitions of 'gender' vary, the term might be appropriate for non-human animals under certain circumstances, such as when the influence of social interactions is a main point of interest (*Cortes et al., 2019*). Operational definitions, even for the term 'sex', are important and, in our experience conducting this study, almost never included in publications. As others have done (e.g., *Duchesne et al., 2020*; *Cortes et al., 2019*; *Holmes and Monks, 2019*; *Johnson et al., 2009*), we emphasize the importance of clear operational definitions while recognizing the limitations of binary categories.

## Limitations of this study

The categorization of each article into a particular discipline was defined exclusively by the journal in which it appeared, in order to be consistent with the original categorizations of *Beery and Zucker, 2011*, and *Woitowich et al., 2020*. For most disciplines, fewer than a dozen articles were in our starting sample; for Neuroscience and Reproduction, only nine. As a result, after we coded the articles, some categories contained few or no articles in a given discipline (see *Supplementary file 1c*). The within-discipline analyses, particularly the pie charts in *Figure 3B*, should therefore be interpreted with caution. Firm conclusions about whether a particular practice is more prevalent in one discipline than another cannot be drawn from the data presented here.

As is the case for any analysis, qualitative or otherwise, our coding was based on our interpretation of the data presentation and wording in the articles. Details of the statistical approach were sometimes left out, leaving the author's intentions ambiguous. Although our approach was as systematic as possible, a small number of articles may have been coded in a way that did not completely capture those intentions. We believe our sample size, particularly in the overall analyses across disciplines, was sufficient to reveal the important trends.

## Conclusion

SABV has been hailed as a game-changing policy that is already bringing previously ignored sex-specific factors to light, particularly for females. In this study, we have shown that a substantial proportion of claimed sex differences, particularly sex-specific effects of experimental manipulations, are not supported by sufficient statistical evidence. Although only a minority of studies that include both sexes actually report data by sex (*Woitowich et al., 2020*), our findings suggest that when data *are* reported by sex, critical statistical analyses are often missing and the findings likely to be interpreted in misleading ways. Note that in most cases, our findings do *not* indicate that the conclusions were inaccurate; they may have been supported by appropriate statistical analyses. Our results emphasize the need for resources and training, particularly those relevant to the study designs and analyses that are commonly used to detect sex differences. Such training would benefit not only the researchers doing the work, but also the peer reviewers, journal editors, and program officers who have the power to hold researchers to a higher standard. Without better awareness of what can and cannot be concluded from separate analysis of males and females, SABV may have the undesired effect of reducing, rather than enhancing, rigor and reproducibility.

## Materials and methods

We conducted our analysis using journal articles from a list published by *Woitowich et al., 2020*. In their study, which was itself based on a study by *Beery and Zucker, 2011*, the authors selected 720 articles from 34 journals in nine biological disciplines. The disciplines were defined by Beery and Zucker and were each represented by four journals, with the exception of Reproduction, which was represented by two (*Table 1*). To be included, articles needed to be primary research articles not part of a special issue, describe studies conducted on mammals, and be published in English. For each journal, Woitowich et al. selected the first 20 articles meeting these criteria published in 2019 (40 articles for Reproduction). For most disciplines, all articles were published between January and April, 2019; for others, articles could have been published as late as June, August, or October for Endocrinology, Behavioral Physiology, and Behavior, respectively.

*Woitowich et al., 2020*, coded each article with respect to whether it contained data analyzed by sex, defined as either that the sexes were kept separate throughout the analysis or that sex was included as a fixed factor or covariate. Of the original 720 articles analyzed, 151 met this criterion. We began our study with this list of 151 articles. Four articles were excluded because they contained data from only one sex, with animals of the other sex used as stimulus animals or to calculate sex ratios.

Our coding strategy was collaborative (*Saldana, 2021*). The majority of the articles (n = 131 out of 151) were read by the first author (YGS) to ascertain the basic experimental designs in the dataset. A subset of the articles (n = 34), spanning a variety of designs, was then discussed between the authors to develop an analysis strategy. This strategy consisted of decision trees used to assign articles to hierarchical categories pertaining to each of four central questions (see below). Once the authors agreed on a set of categories that would effectively capture the variables of interest, the second author (DLM) coded all of the articles, assigning each to one category per question. During coding, articles for which the most appropriate category was not immediately obvious were discussed between the authors until agreement was reached. This process resulted in the modification of some of the categories, which improved conceptual clarity and strengthened the analysis. Once the decision tree was finalized (*Supplementary file 1a*) and all articles were coded accordingly, the first author then independently coded three articles from each discipline to assess reliability of the method (Q1-Q4 for each of 27 articles). Interrater reliability, calculated as the number of agreements divided by the number of agreements plus disagreements, was 93%. During the subsequent discussion, the categorization was changed to that of the first author for approximately half of the discrepancies (3% of the total number of decisions); the other 3% remained the same. The final categorizations in *Supplementary file 1b* represent consensus between the authors after all readings and discussions.

### Question 1: Was a sex difference reported?

Because we were interested in the frequency with which sex differences were found, we first identified articles in which the sexes were explicitly compared. We counted as a comparison any of the following: (1) sex was a fixed factor in a statistical model; (2) sex was included as a covariate in a statistical model *and* a p value for the effect of sex was reported; (3) a p value for a comparison of means between males and females was presented; (4) the article contained wording suggestive of a comparison, for example, 'males were larger than females'. We also included articles with wording suggestive of a sex difference in response to a treatment, for example, 'the treatment affected males *but not* females' or 'the males responded to treatment, *whereas* the females did not', or 'the treatment had a *sex-specific* effect'. Similarly, we included here articles with language referring to a non-difference, for example, 'we detected no sex differences in size' or 'the response to treatment was similar in males and females'. Articles in which sex was included as a covariate for the purposes of controlling for sex, rather than comparing the sexes, were not coded as having compared the sexes (see *Beltz et al., 2019*). When the sexes were compared but no results of those comparisons, for example, p values, were reported, that omission was noted and the article was coded accordingly. Each article in which the sexes were compared was then further coded as either reporting a sex difference or not, and if so, whether a sex difference was mentioned in the title or abstract. If mentioned in the title or abstract, the sex difference was coded as a 'major finding'; otherwise, sex differences mentioned in the body of the paper, figures, or tables were coded as 'minor'.

## Question 2: Did the article contain a study with a factorial design?

We looked for studies with a 2 × 2 factorial design (*Figure 2A*) in which sex was one of the factors. Sex did not need to be explicitly identified as a fixed factor; we included here all studies comparing across levels of one factor that comprised females and males with each of those levels. In some cases that factor was a manipulation, such as a drug treatment or a gene knockout. Non-sex factors also included variables such as age, season, presentation of a stimulus, etc. For simplicity, we refer to the other factor as 'treatment'. Any article containing at least one such study was coded as having a factorial design. The other articles were coded as containing no comparisons across sex or as containing only group comparisons across sex. The latter category included studies with sex as a covariate of interest in a model such as a multiple regression, if the authors were not making any claims about potential interactions between sex and other variables.

For studies with a factorial design, we further coded the authors' strategy of data analysis. First, we noted whether authors tested for an interaction between sex and treatment; that is, they tested whether the effect of treatment depended on sex. We coded as 'yes' one study in which the magnitude of the differences between treated and control groups was explicitly compared across sex. For articles containing tests for interactions, we noted the outcome of that test and the interpretation. Articles containing no tests for interactions were assigned to one of several sub-categories in the following order (coded as the first category on this list for which the description was met for any analysis in the article): tested for effects of treatment within sex, tested for effects of sex within at least one level of treatment, or tested for main effects of sex only. Within each of those categories we further coded the outcome/interpretation, for example, sex difference or no sex difference. Any articles containing statements that the sexes responded differently to treatment or that the response was 'sex-specific' were coded as reporting a sex-specific effect. We also noted when authors reported an absence of such a result. Articles not comparing across sex at all, with statistical evidence or by assertion, were coded accordingly.

## Question 3: Did the authors pool males and females?

We assigned articles to one of three categories: analyzed males and females separately throughout, included sex in the statistical model for at least some analyses (with the rest analyzed separately), or pooled for at least some analyses. The second category, included sex in the model, included articles in which AIC or similar statistic was used to choose among models that included sex, although sex may not have been in the model ultimately chosen. This category did not distinguish between analyses including sex as a fixed factor vs. a covariate; this distinction is noted where relevant in *Supplementary file 1b*. Any article containing pooled data was coded as pooled, even if some analyses were conducted separately or with sex in the model. For articles that pooled, we further noted whether the authors tested for a sex difference before pooling and, if so, whether p values or effect sizes were reported.

## Question 4: Did the authors use the term 'sex' or 'gender'?

We searched the articles for the terms 'sex' and 'gender' and noted whether the authors used one or the other, both, or neither. Terms such as 'sex hormones' or 'gender role', which did not refer to sex/gender variables in the study, were excluded from this assessment. For the articles using 'gender', we further noted when the term was used for non-human animals.

To visualize the data, we used river plots (*Weiner, 2017*), stacked bar graphs, and pie charts based on formulae and data presented in *Supplementary file 1c*.

## Acknowledgements

We thank Nicole Baran, Isabel Fraccaroli, and Naomi Green for assistance and suggestions in the initial stages of this project, and Chris Goode for assistance with the river plots. We are grateful to Lise Eliot, Chris Goode, Niki Woitowich, Colby Vorland, Chanaka Kahathuduwa, and an anonymous reviewer for providing comments on the manuscript.

## Additional information

### Funding

| Funder | Grant reference number | Author |
| --- | --- | --- |
| Emory University Research Committee | 00106050 - URC 2021-22 | Donna L Maney |

The funders had no role in study design, data collection and interpretation, or the decision to submit the work for publication.

### Author contributions

Yesenia Garcia-Sifuentes, Data curation, Investigation, Project administration, Supervision, Validation, Visualization, Writing – original draft, Writing – review and editing; Donna L Maney, Conceptualization, Data curation, Formal analysis, Funding acquisition, Investigation, Methodology, Supervision, Validation, Visualization, Writing – original draft, Writing – review and editing

### Author ORCIDs

Yesenia Garcia-Sifuentes ⓘ http://orcid.org/0000-0001-8918-304X
Donna L Maney ⓘ http://orcid.org/0000-0002-1006-2358

### Decision letter and Author response

Decision letter https://doi.org/10.7554/70817.sa1
Author response https://doi.org/10.7554/70817.sa2

## Additional files

### Supplementary files

• Supplementary file 1. Codes and data for the bibliometric analysis. (1 a) Articles were coded into the categories shown for each of four questions. (1b) Codes for all articles are indicated in the 'Question' columns. See (a) for the explanations of the codes. All codes marked with an asterisk are explained in the 'notes' column. Columns D and E show the title and date of publication for each article, which was also shown in *Woitowich et al., 2020*. (1 c) Numbers of articles coded into each category. Column A shows the total number of articles; columns D-L show them broken down by discipline.

• Transparent reporting form

### Data availability

All data generated or analysed during this study are included in the manuscript and supporting files.

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
