## [Decision Letter]

**Decision letter after peer review:**

Thank you for submitting your article "Reporting and Misreporting of Sex Differences in the Biological Sciences" for consideration by *eLife*. Your article has been reviewed by 3 peer reviewers, and the evaluation has been overseen by a Reviewing Editor and Mone Zaidi as the Senior Editor. The following individuals involved in review of your submission have agreed to reveal their identity: Colby J Vorland (Reviewer #1); Chanaka Kahathuduwa (Reviewer #3).

*Reviewer #1 (Recommendations for the authors):*

Regarding the coding of papers, additional details about the subset of articles independently coded would be helpful to assess how you are confidence in the extraction quality.

Thank you for providing your data along with the manuscript. I have no further comments separate from my public review.

*Reviewer #2 (Recommendations for the authors):*

It seems like a severe omission in the introduction to not include and discuss the NIH's own retrospective on their requirement:

Arnegard, M. E., Whitten, L. A., Hunter, C., and Clayton, J. A. (2020). Sex as a Biological Variable: A 5-Year Progress Report and Call to Action. Journal of women's health (2002), 29(6), 858-864. https://doi.org/10.1089/jwh.2019.8247

The operationalization of a "major finding" is not defined in the text, just buried in a figure caption.

River plots should have counts/percentages included with them.

Figure 1B – Why are the percentages not out of 100%? If the remainders are "Sexes not compared" then that should be included as a fourth category in the plot.

Figures 2D-F – Same concern as with 1B.

The difference between Figures 2D and 2E is not well explained.

"We found that a large majority of studies on non-human animals used "sex" to refer to the categorical variable comprising females and males (Figure 4)." Are the counts in Figure 4 for the subset of non-human studies? The description in the Results section and figure caption do not mention that earlier.

"…we have shown that a substantial proportion of claimed sex differences, particularly sex-specific effects of experimental manipulations, are not supported by statistical evidence." This refers to my earlier comment. Ironically given the subject matter, I don't feel like a single proportion was given to this characterization of articles.

*Reviewer #3 (Recommendations for the authors):*

1. Please describe the methods used to classify the articles by discipline. If this method was not objective, consider discussing that as a limitation.

2. The studies were coded only by one author (author 2) and this may compromise the validity of the findings. To mitigate this limitation, a second coder could rate all the studies (or at least a random sample of the studies) using the same criteria and the inter-rater reliability could be presented. If this is not possible, the impact of this major limitation on the conclusions needs to be discussed in the manuscript.

3. Please consider revising the following statements:

a. "our finding could indicate that researchers interested in sex differences are primarily the ones following NIH guidelines." – this statement is not supported by the data.

b. "The prevalence of this issue is difficult to estimate because opinions vary about when corrections are necessary" (regarding correcting for multiple comparisons) – the prevalence of the problem could be objectively estimated irrespective of one's opinion about the need to adjust to conserve FWER.

c. "This study was underpowered for examining these issues within any particular discipline" – since statistical tests were not performed, it is best to avoid the term "underpowered".

4. Please consider presenting the proportions and / or percentages in the river plots. This will substantially facilitate comprehension of the results.

5. Please consider reporting the reference of each article in Table S2 to enhance transparency and reproducibility. It is not possible to track most of the reviewed articles using the information presented in the manuscript or the supplementary material.

---

## [Author Response]

The reviewers have discussed their reviews with one another, and the Reviewing Editor has drafted this to help you prepare a revised submission.Reviewer #1 (Recommendations for the authors):Regarding the coding of papers, additional details about the subset of articles independently coded would be helpful to assess how you are confidence in the extraction quality.

We have expanded our description of how coding was handled between the two authors. Interrater reliability is now presented and was above 90%.

Thank you for providing your data along with the manuscript. I have no further comments separate from my public review.Reviewer #2 (Recommendations for the authors):It seems like a severe omission in the introduction to not include and discuss the NIH's own retrospective on their requirement:Arnegard, M. E., Whitten, L. A., Hunter, C., and Clayton, J. A. (2020). Sex as a Biological Variable: A 5-Year Progress Report and Call to Action. Journal of women's health (2002), 29(6), 858-864. https://doi.org/10.1089/jwh.2019.8247

This article is now cited and a few of its major points are covered in the first paragraph of the manuscript.

The operationalization of a "major finding" is not defined in the text, just buried in a figure caption.

Major vs. minor finding was defined in the first paragraph under “Question 1” (formerly lines 88-91). The relevant sections of the Results and Methods have been rewritten so that this definition is more explicitly stated.

River plots should have counts/percentages included with them.

The counts have been added to both river plots.

Figure 1B – Why are the percentages not out of 100%? If the remainders are "Sexes not compared" then that should be included as a fourth category in the plot.

“Sexes not compared” has been added to Figure 1B to bring the totals in each column up to 100%.

Figures 2D-F – Same concern as with 1B.

Figure 2E (formerly 2F) already goes to 100% in both columns. Figures 2D and 2F (formerly 2E) plot the percentages of papers with a factorial design and the percentages making claims of a sex-specific-effect, respectively. The latter two categories are then broken down into whether the authors tested for or reported an interaction, which can be done only for the articles that are actually in the graph. It would not be appropriate, for example, to test for an interaction when the design is not factorial, so those papers do not appear in Figure 2D.

The difference between Figures 2D and 2E is not well explained.

We have clarified in the figure caption what is shown in each panel. We have also swapped the order of Figures 2E and 2F, since 2D was more closely related to what is now 2E.

"We found that a large majority of studies on non-human animals used "sex" to refer to the categorical variable comprising females and males (Figure 4)." Are the counts in Figure 4 for the subset of non-human studies? The description in the Results section and figure caption do not mention that earlier.

The counts in Figure 4 pertain to all 151 of the papers in the study, not just the non-human ones. This has been clarified in the caption. The reference to Figure 4 has been deleted in the sentence the reviewer is referring to – we thank the reviewer for pointing out that the figure did not contain information relevant to the point being made there. We have added to the Results the proportion of articles on non-humans that used “sex”.

"…we have shown that a substantial proportion of claimed sex differences, particularly sex-specific effects of experimental manipulations, are not supported by statistical evidence." This refers to my earlier comment. Ironically given the subject matter, I don't feel like a single proportion was given to this characterization of articles.

The reviewer is asking for two different statistics. First, they pointed out above that we “did not report overall percentages of articles that "did it right" (e.g., the original authors appropriately reported on and correctly interpreted the results of an interaction).” This statistic, in other words the proportion of articles that did it right, was originally reported in the first paragraph about Question 2 in the Results. A sentence has been added to reiterate and clarify this statement. The same point was also in the Discussion, although there we gave only the percentage that did it “wrong”. The number that did it right (% wrong subtracted from 1) is now also included there.

Second, the reviewer wants to know, out of the studies that claimed sex-specific effects, what proportion actually showed statistical evidence for such effects. This statistic was given in Figure 2F (now Figure 2E), the first column, which shows that ~30% of the articles reporting sex-specific effects tested for such. The statistic was also given in the text, under Question 2. References to the relevant figure have been added to these sentences.

Reviewer #3 (Recommendations for the authors):1. Please describe the methods used to classify the articles by discipline. If this method was not objective, consider discussing that as a limitation.

Articles were categorized according to the journal in which they appeared. We followed the categorization used by Beery et al., (2011) and Woitowich et al., (2020) for the sake of consistency with those studies. This has been clarified in Table 1 and in the Methods.

2. The studies were coded only by one author (author 2) and this may compromise the validity of the findings. To mitigate this limitation, a second coder could rate all the studies (or at least a random sample of the studies) using the same criteria and the inter-rater reliability could be presented. If this is not possible, the impact of this major limitation on the conclusions needs to be discussed in the manuscript.

As noted above in our response to R1, we have expanded our description in the Methods of how coding was handled among the two reviewers and how we are confident in each decision. We have calculated interrater reliability as greater than 90%.

3. Please consider revising the following statements:a. "our finding could indicate that researchers interested in sex differences are primarily the ones following NIH guidelines." – this statement is not supported by the data.

We found that in the subset of studies in which data were disaggregated, in other words

NIH guidelines were followed, the sexes were usually also compared. There are two

possible explanations for this finding – first, researchers who follow NIH guidelines went a step beyond to compare the sexes; second, that researchers interested in sex

differences are primarily the ones following the guidelines. These are slightly different

conclusions and our data do not distinguish between them. We have rephrased this

statement so that we do not claim that our data led us to any particular conclusion.

b. "The prevalence of this issue is difficult to estimate because opinions vary about when corrections are necessary" (regarding correcting for multiple comparisons) – the prevalence of the problem could be objectively estimated irrespective of one's opinion about the need to adjust to conserve FWER.

Any estimation of the prevalence of an error requires a clear definition of the error. Without such a definition, it is not possible to categorize articles as having committed vs. not committed the error. In this case, the conditions under which corrections are necessary, and the type of correction that is needed, are extremely variable. For example, some researchers argue that a correction is necessary even when sex is included in an omnibus test; others argue that the omnibus test itself obviates the need for corrections. Some researchers argue that corrections are necessary when multiple tests are done in the same set of animals, others insist that the correction must be applied to all experiments in the same manuscript even when separate samples are analyzed. Some point out that in order to implement a correction logically and consistently it must be applied over all studies for a PI’s entire career, so it is therefore overkill in all cases and should never be done. As we do not wish to claim authority regarding what is acceptable or not in this particular area, we decided not to categorize articles according to whether they corrected for multiple comparisons in an acceptable way. Instead, we mention in the Discussion a few extreme cases in order to call attention to the issue. The sentence the reviewer referred to has been omitted in the revision.

c. "This study was underpowered for examining these issues within any particular discipline" – since statistical tests were not performed, it is best to avoid the term "underpowered".

This sentence has been removed.

4. Please consider presenting the proportions and / or percentages in the river plots. This will substantially facilitate comprehension of the results.

The counts have been added to both river plots.

5. Please consider reporting the reference of each article in Table S2 to enhance transparency and reproducibility. It is not possible to track most of the reviewed articles using the information presented in the manuscript or the supplementary material.

A column has been added to the table for the title of the article, to follow the precedent set by Woitowich et al. (2020).